Extended exposure to elevated temperature affects escape response behaviour in coral reef fishes

Warren Donald T. donald.warren@my.jcu.edu.au 1 2
Donelson Jennifer M. 2 3
McCormick Mark I. 1 2
1 Department of Marine Biology and Aquaculture, James Cook University , Townsville , Queensland , Australia
2 ARC Centre of Excellence for Coral Reef Studies, James Cook University , Townsville , Queensland , Australia
3 School of Life Sciences, University of Technology Sydney , Sydney , New South Wales , Australia
Hedrick Ann
Electronic publication date: 2017 Aug 18
Publication date: 2017
Volume: 5
Electronic Location ID: e3652
Received 2016 Nov 13; Accepted 2017 Jul 14
Copyright: ©2017 Warren et al.
Copyright year: 2017
Copyright holder: Warren et al.
License: This is an open access article distributed under the terms of the Creative Commons Attribution License, which permits unrestricted use, distribution, reproduction and adaptation in any medium and for any purpose provided that it is properly attributed. For attribution, the original author(s), title, publication source (PeerJ) and either DOI or URL of the article must be cited.
License URL: https://creativecommons.org/licenses/by/4.0/

Keywords: Temperature, Fast-start response, Climate change, Coral reef fish, Predator–prey

Funding: Postgraduate Research Scholarship by James Cook University (DTW) University of Technology Sydney and The Ian Potter Foundation (JMD) ARC Centre of Excellence for Coral Reef Studies and the College of Marine and Environmental Sciences at James Cook University (MIM) DTW was supported through a Postgraduate Research Scholarship by James Cook University. JMD was supported by the University of Technology Sydney and The Ian Potter Foundation. MIM was supported by the ARC Centre of Excellence for Coral Reef Studies and the College of Marine and Environmental Sciences at James Cook University. The funders had no role in study design, data collection and analysis, decision to publish, or preparation of the manuscript.

==============================
The threat of predation, and the prey’s response, are important drivers of community dynamics. Yet environmental temperature can have a significant effect on predation avoidance techniques such as fast-start performance observed in marine fishes. While it is known that temperature increases can influence performance and behaviour in the short-term, little is known about how species respond to extended exposure during development. We produced a startle response in two species of damselfish, the lemon damsel Pomacentrus moluccensis, and the Ambon damselfish Pomacentrus amboinensis, by the repeated use of a drop stimulus. We show that the length of thermal exposure of juveniles to elevated temperature significantly affects this escape responses. Short-term (4d) exposure to warmer temperature affected directionality and responsiveness for both species. After long-term (90d) exposure, only P. moluccensis showed beneficial plasticity, with directionality returning to control levels. Responsiveness also decreased in both species, possibly to compensate for higher temperatures. There was no effect of temperature or length of exposure on latency to react, maximum swimming speed, or escape distance suggesting that the physical ability to escape was maintained. Evidence suggests that elevated temperature may impact some fish species through its effect on the behavioural responses while under threat rather than having a direct influence on their physical ability to perform an effective escape response.

Introduction

Predator–prey interactions are one of the key drivers structuring communities (Lima & Dill, 1990). Predation pressure can alter prey morphology, regulate population size, and control biodiversity (Sih et al., 1985). Similarly, antipredator responses of prey reduce the efficacy of predators and enhance survival. Factors that influence predator attack success or prey escape performance may alter the outcome of these interactions, potentially causing a cascade of trophic effects. Increased temperature has already been shown to affect antipredator responses in terrestrial ectotherms (Cury de Barros et al., 2010), freshwater fishes (Weetman, Atkinson & Chubb, 1998), and marine species (Allan et al., 2015). Understanding how changes in environmental temperature can affect key ecological processes can be useful when predicting the composition of future communities.

Rising environmental temperatures can substantially impact the behaviour and locomotor performance components of escape responses in fish (Domenici, 2010). Previous studies have shown that temperature can influence locomotor performance through physiological mechanisms like muscle performance (Johnston, Fleming & Crockford, 1990) and enzyme activity (Johnson & Bennett, 1995). Additionally, temperature can affect behavioural components of escape responses, including responsiveness (Preuss & Faber, 2003) and directionality (Szabo et al., 2008). The effect of environmental temperature will depend on both the level of thermal change compared to the normal thermal conditions, and the length of time thermal change is experienced (Munday, McCormick & Nilsson, 2012). With sufficient exposure time, plasticity can occur resulting in changes to the phenotype, which potentially reduce the negative impacts of rising temperature (Angilletta, 2009). Exposure during the early stages of life is believed to unlock greater plasticity in traits (West-Eberhard, 2003), and this has been found to be true for coral reef fishes (Donelson et al., 2011). While many studies have investigated the effects of short-term exposure to elevated temperatures in adult fishes, to date few studies have examined how juvenile fishes respond to long-term exposure to near-future temperature increases.

The interaction between a predator and its prey is an intricate sequence that, at its end, leads to a predator striking at a prey and the prey being captured or escaping (Lima & Dill, 1990). Many amphibians and fishes undertake a C-start escape response to a startling stimulus (Bullock, 1984) involving a sudden acceleration away from the direction of the perceived threat (Domenici & Blake, 1997). C-start responses are controlled by two large neurons called Mauthner cells, one for each lateral half of the body (Eaton, Bombardieri & Meyer, 1977). When a stimulus is detected, the corresponding Mauthner cell triggers the escape response resulting in a rapid contraction of white muscle fibres. At first, the fish bends into a C-shape (stage 1), then contralateral contraction releases this tension which propels the fish to flee the stimulus in the opposite direction (stage 2) (Eaton, Bombardieri & Meyer, 1977). The initiation of C-starts are neurally controlled with maximum performance believed to be limited by muscle power output (Wakeling & Johnston, 1998). However, fish rarely perform at their maximal limits (Webb, 1986) and modulation of performance during an escape response has been shown to occur in different behavioural contexts, showing there is potential for optimisation (Korn & Faber, 2005; Domenici, 2010; Ramasamy, Allan & McCormick, 2015). This supports the economic model that individuals balance the energetic cost of escaping with the threat of predation (Ydenberc & Dill, 1986). How long-term exposure to elevated temperature can alter behavioural components of an escape response is still unknown.

The present study examines whether short-term or long-term exposure to elevated temperature regimes affect the fast-start escape response in two tropical damselfishes. Using a repeatable drop stimulus (i.e., a stimulus that was consistently replicated), we compared escape responses of juveniles from current-day controls (29 °C) to individuals exposed to 50 and 100 year projections (30 and 31 °C; (Collins et al., 2013) for either 4d or 90d. Specifically, we tested: (1) the effects of short-term exposure to elevated temperature as an indication of thermal sensitivity and (2) how short-term exposure differed to long-term exposure at elevated temperature as an indication of plasticity. The inclusion of two species was to explore the species-specific effect that temperature had on escape responses.

Methods

Ethics statement and collecting permits

All work reported herein was conducted under permits from James Cook University Animal Ethics Committee (A2079), the Great Barrier Reef Marine Park Authority (G10/33239.1), and Queensland Fisheries (170251).

Study species, collection, and holding facilities

To examine how short-term and long-term exposure to elevated temperatures influences escape performance, two reef fish species were used (the Lemon damselfish, Pomacentrus moluccensis, and the Ambon damselfish, Pomacentrus amboinensis). These species have been used in past studies on escape performance (Allan et al., 2013) and behaviour (McCormick & Weaver, 2012) making them familiar study species. These species co-occur across the Indo-Pacific from the southern Great Barrier Reef up to Indonesia and Japan (Allen, 1991). Like many reef fish, these species undergo embryonic development during a pelagic larval phase. After 3–4 weeks, larvae make their final metamorphosis and recruit back to the reef to join the adult population. Both species settle to the reef at similar ages, though standard length of P. amboinensis, tends to be slightly larger (McCormick & Weaver, 2012). Here, thermal plasticity is ecologically important as individuals often settle to reefs with thermal regimes different to what they experienced as larvae. Recently settled recruits (aged 1–2 months) were collected from the Cairns region (−16°78′S, 146°26′E) of the Great Barrier Reef, Australia in January 2014. At this latitude, these species do not mature for at least one year, making our individuals immature juveniles. Therefore, gender was not observed as it is not relevant at this age. Once mature, individuals will sex-change depending on social structure (Fishelson, 1998).

Fish were transported back to a closed system holding facilities at the Marine and Aquaculture Research Facilities Unit (MARFU), James Cook University, Townsville. Fish were housed in densities of three of varying sizes (e.g., one big, medium, and small) in 40 litre replicate tanks. This sizing order helped formed a natural hierarchy within tanks and reduced within tank fighting. Tanks were randomly allocated into one of three temperatures: 29.0 °C (current-day summer average for the collection region; control; (AIMS, 2014), 30.0 °C, or 31.0 °C (projected 50 and 100 year future temperatures; Collins et al., 2013). Holding tank temperatures were computer monitored via Innotech Genesis II controller V5 temperature system capable of controlling random fluctuation less than 0.1 °C for the system. The system was continuously supervised by the in-house staff at the facility and automatically notified of any irregularities. To avoid thermal stress, the temperature was raised 1 °C per day (Gardiner, Munday & Nilsson, 2010; Nilsson, Östlund-Nilsson & Munday, 2010). Elevated temperatures were split into two exposure lengths: 4d or 90d. A 90d period was chosen for our long-term exposure treatment based on previous developmental studies on this species (Grenchik, Donelson & Munday, 2013) and a closely related species of damselfish (Donelson et al., 2011). A 4d period was chosen for our short-term exposure treatment to observe the effects of increased temperature without causing a thermal stress response (Gardiner, Munday & Nilsson, 2010). No thermal benefit in plasticity has been shown to occur in P. moluccensis for up to 22 days of exposure to similar elevated temperatures supporting our 4d treatment would not be confounded by reversible plasticity (Nilsson, Östlund-Nilsson & Munday, 2010). Temperature variation and photoperiod followed natural diurnal cycle for the collection region, ±0.6 °C around the mean and 12:12 h respectively (AIMS, 2014). Food quantity was not controlled for and all fish were given a high performance commercial fish pellet (NRD 0.5–0.8 mm; Primo Aquaculture, QLD, Australia) once or twice daily to satiation. Excess food was removed and tanks were cleaned via a vacuum siphon weekly or as needed.

Experimental design

The escape performance of fishes was quantified using a repeatable drop stimulus (i.e., a stimulus with mechanical release that could be replicated consistently). Thermal sensitivity to elevated temperature (aim 1) was measured by exposing individuals to either 30 or 31 °C for a 4d period. To explore the extent of thermal plasticity (aim 2), separate treatments were held at either 30 or 31 °C for 90d. The control treatment remained at 29 °C for the entirety of the experiment. In total, there were 10 treatments: two short-term exposures, two long-term, one control, and repeated for each species. Holding tank temperatures were raised 1 °C/day to reduce the effect of heat shock (Gardiner, Munday & Nilsson, 2010). To control for any effect of time spent in captivity, the 4d treatments were first held at control temperature, then exposed to their randomly assigned temperature four days before the end of the long-term duration. This allowed all fish to finish temperature treatments at the same time.

Experimental procedure

After the exposure period, individuals were measured (standard length, x¯±SD; P. amboinensis n = 58, 30.57 ± 5.79 mm, P. moluccensis n = 83, 28.08 ± 3.59 mm) and tested for escape performance the following day. Treatments of P. amboinensis were significantly larger than P. moluccensis (F1,131 = 9.52, p = 0.002), however there were no differences across treatments for each species. Escape performance was tested in a circular arena (30 cm diameter) filled to a depth of 10 cm to reduce movement in the vertical plane and illuminated with fluorescent lighting strips. Water temperature of the arena reflected the treatment temperature that fish were held in. Water temperature was monitored with a C20 Comark thermometer (Comark Corporation, Norfolk, UK). Temperature was maintained with a glass bar heater (Aqua One, 300 W) and air stone for aeration. Complete water exchanges were conducted every 10 min or after 3 fish. The walls and top had opaque covers to prevent outside disturbance. To elicit an escape response, a startling stimulus consisting of a weight with a tapered end was released through a tube suspended above the centre of the arena. The suspended tube stopped ∼1 cm above the water’s surface concealing the stimulus weight as it fell thus creating a more sudden disturbance. A monofilament line was used to raise the weight and was long enough that the end touched just the water’s surface when released, preventing collision with the individual. To start a trial, fish were placed into a central habituation ring (10 cm diameter) via a water filled sample jar and given a one-minute habituation period. Following the habituation period, the central ring was gently raised and the stimulus weight was released, striking the water in an attempt to elicit an escape response. If a C-start was performed, no further attempts were conducted and the fish was returned to its holding tank. If no reaction was observed, fish were replaced into the habituation ring for another minute and the second attempt was conducted. Fish were given three attempts to perform a C-start before being recorded as “no reaction”. The escape response was recorded using a high-speed camera (Casio Exilim Ex-F1; 600 fps) directed at a 45° angled mirror placed beneath the testing tank.

Response variables

Response variables were measured using the fish’s centre of mass (∼35% SL from the snout; Webb, 1976) and tracked using ImageJ software (v1.48) with the manual tracking plugin. We chose our variables as they have been used in previous study as a good indicator of escape ability (Walker et al., 2005). Only stages 1 and 2 of the C-start, described as directional changes in the anterior part of the body (Domenici & Blake, 1997) were quantified.

Non-locomotor

(a) Responsiveness: proportions of fish that responded with a “C-start” or “no C-start”. No C-start was subdivided into “avoid” to the stimulus by swimming away but not performing a desired C-start and “no reaction” to the stimulus.

(b) Directionality: left and right side were split anteroposteriorly from directly above the fish. Directionality was determined by whether the head turned “away” or “toward” the side the stimulus occurred during stage 1 of the response.

(c) Response latency (ms): time between the stimulus onset and first movement of the individual.

Locomotor

(d) Escape distance (mm): distance travelled during the response to the end of stage 2.

(e) Maximum swim speed (body lengths s−1): maximum velocity achieved at any time during the escape response.

Data analysis

Statistical analyses were performed using IBM SPSS statistics (v23.0.0.2). Proximity of fish to the stimulus at the onset and body size can affect escape responses (Webb, 1976). The central habituation ring helped reduce differences in starting proximity to stimulus and the remaining distances of fish to the stimulus did not differ among treatments (one-way ANOVA F4,110 = 0.959, P = 0.433). Maximum swim speed (m s−1) was converted to body length s−1 to control for any effect of size. Control temperature did not consist of two exposure combinations like the elevated temperatures, therefore a full-factorial 3 × 2 design was not possible. Instead, the five treatments (control, two short-term, and two long-term) were categorised into one “temperature treatment” factor. A binomial logistic regression was used to ascertain the effects of temperature treatment on the likelihood of an individual to elicit a C-start after the stimulus and the likelihood of that individual to turn away from the stimulus. Response latency, maximum swim speed, and escape distance were used as dependent variables analysed separately with one-way ANOVAs (α = 0.05) and temperature treatment as the fixed factor. These analyses were repeated for both species (replicates ranged from 7–12 for P. amboinensis and 9–19 for P. moluccensis). Preliminary analysis of response latency, maximum swim speed, and response distance used a linear mixed effects model with holding tank as a random factor to test for a tank effect. However, AIC values were lower for models not including random tank factor with ΔAIC <2 to models that did include random tank factor. Consequently, this term was dropped from the final model. We predicted that short-term exposure to elevated temperature would have a negative impact on escape responses. If beneficial plasticity was possible with long-term exposure, individuals exposed for 90d would exhibit improvement in performance that either approached or returned to present-day control levels.

Results

There was no significance in the logistic regression model for responsiveness in P. amboinensis (χ2(4) = 4.18, p = 0.38; Fig. 1A). At control, 84.62% of individuals responded with a C-start, 7.69% showed no reaction, and 7.69% displayed avoidance behaviour. Exposure of four days to either 30 or 31 °C had very little effect on responsiveness. However, for individuals that did not display a C-start, all showed no reaction to the stimulus and none displayed avoidance behaviour. After 90 days, fish were 2.19 times less likely (71.43%) to perform a C-start at 30 °C and 3.14 times less likely (63.64%) at 31 °C. Additionally, avoidance behaviour was returned (14.29%) and divided non C-start individuals in half with those showing no reaction (14.29%). There was significance in the regression model for directionality in P. amboinensis (χ2(4) = 11.23, p = 0.024; Fig. 1B). The model explained 28.70% (Nagelkerke R2) of the variance in directionality of C-starts and correctly classified 67.60% of cases. Compared to controls, fish exposed to 30 °C for 4 days were 5 times more likely (33.33%) to turn towards the stimulus and 6.67 times more likely (40%) after 90 days. At 31 °C, fish were 25 times more likely (71.43%) to turn towards the stimulus, which was the same for both exposure durations. Temperature nor exposure duration had any significant effect on response latency (F4,42 = 1.32, p = 0.27, partial η2 = 0.053; Fig. 2A), maximum swim speed (F4,41 = 0.6, p = 0.66 partial η2 = 0.056; Fig. 2B), or response distance (F4,42 = 0.65, p = 0.62, partial η2 = 0.058; Fig. 2C) for P. amboinensis.

Figure 1 Responsiveness and directionality to the startling stimulus.

Percent of reactions represents C-start responsiveness of juvenile Pomacentrus amboinensis (A) and Pomacentrus moluccensis (C) to a drop stimulus at control, 4d or 90d exposure durations to elevated temperatures. Types of reaction are: no reaction (grey), avoidance (open), and C-start (solid). Directionality categorises only C-start individuals by the percent of turns made by P. amboinensis (B) and P. moluccensis (D) where the first movement of head was either towards (solid) or away (open) from the stimulus.

Figure 2 Measures of kinematic variables to the drop stimulus.

Kinematic performance of fast startsby juvenile P. amboinensis and P. moluccensis at control (solid), 4d (open) or 90d (grey) exposure durations to elevated temperatures. Variables measured were response latency, max speed (body lengths s −1, and escape distance (A, B, and C respectively for P. amboinensis and D, E, and F respectively for P. moluccensis).

For P. moluccensis, logistic regression model for responsiveness was statistically significant (χ2(4) = 16.58, p = 0.002; Fig. 1A). The model explained 28% (Nagelkerke R2) of the variance in proportion of individuals that elicited a C-start and correctly classified 80% of cases. At control, 82.61% of fish displayed a C-start. Of the fish that did not C-start, most showed avoidance behaviour (13.04%) and no reaction (4.35%) least of all. Responsiveness rose to absolute (i.e., 100% of individuals performed a C-start) after four days of exposure to either 30 or 31 °C. When 90 day exposure was compared to controls, individuals were 1.45 times less likely (76.47%) to perform a C-start at 30 °C and 3.55 times less likely (57.14%) at 31 °C. For individuals that did not display a C-start, there was an increase in no reaction up to 38% at 31 °C (from 4.35% at control). There was no significance in the regression model for directionality (χ2(4) = 5.98, p = 0.20; Fig. 1B). Fish with four days of exposure to 30 °C were 2.8 times more likely (50%) to turn towards the stimulus compared to controls and 4.2 time more likely (60%) at 31 °C. However, this was reduced after prolonged exposure where directionality was very similar to control (23.08% and 25% for 30 and 31 °C, respectively). There was no significant effect of temperature and exposure duration on response latency (F4,60 = 1.23, p = 0.30, partial η2 = 0.052; Fig. 2A), maximum swim speed (F4,60 = 0.19, p = 0.94, partial η2 = 0.013; Fig. 2B), nor response distance (F4,60 = 0.85, p = 0.49, partial η2 = 0.054; Fig. 2C) for P. moluccensis.

Discussion

Studying how escape responses are affected by elevated temperature can be important when making predictions on the ability of prey to escape predators under climate change conditions (Gilman et al., 2010). Supporting our predictions, we found short-term exposure to elevated temperature negatively impacted directionality for both species. However, there was little effect on responsiveness, response latency, maximum, swim speed, and escape distance. After long-term exposure, directionality in P. moluccensis followed our predictions by reducing the number of turns towards the stimulus and returning to control levels. For P. moluccensis, long-term exposure also altered responsiveness with fish responding less often with a C-start response at elevated temperatures. Such alterations in the C-start responsiveness to predators is likely to change prey survival, while the differential effects of temperature on the two closely related species suggests that future temperature elevation may affect community composition through differential survival.

The ability for prey to perceive and avoid a predatory threat is essential to survival. Directionality in escape responses is important as turning towards a stimulus would increase exposure time to a predator (Domenici, 2010). Short-term exposure to elevated water temperature negatively impacted directionality by increasing number of turns towards the stimulus. Previous study on goldfish also found directionality was impaired at increased temperatures compared to controls (Szabo et al., 2008). Altered directionality may have been due to an impairment of sensorimotor control (Foreman & Eaton, 1993). While not significant, short-term exposure to elevated temperatures increased C-start responsiveness in P. moluccensis from 80% to 100%, and reduced avoidance reactions to 0% for both species. Generally, proportion of C-starts has been shown to increase with exposure to temperature both above (Szabo et al., 2008) and below control treatments (Preuss & Faber, 2003), suggesting responsiveness may increase with thermal stress, at either extreme. Hyperactivity under stress can be beneficial to survival, as a fish that does not react during a predator attack will most likely get eaten. However, this strategy can risk expending additional energy on a false or non-threatening stimulus.

Extended exposure to new environmental conditions can alter the physiological and behavioural response of individuals (Angilletta, 2009). We found directionality in P. moluccensis showed signs of beneficial plasticity after long-term exposure by returning to control levels. Long-term exposure may have reduced the negative impact of sensorimotor control brought on by thermal stress. Physiological plasticity to similar temperatures (+2 °C) in this species has also been shown to occur for aerobic metabolism (Grenchik, Donelson & Munday, 2013) potentially demonstrating that P. moluccensis could exhibit beneficial plasticity to future projected temperatures in a range of traits. The proportion of C-starts reduced while avoidance behaviour increased with long-term exposure in both species. Decreasing the proportion of C-starts deviates further from control levels, therefore cannot be considered beneficial. However, future sea temperature increases are expected to exceed the thermal optimum for many reef fish (Tewksbury, Huey & Deutsch, 2008), forcing species to adopt energy saving strategies (Ydenberc & Dill, 1986). Reducing the proportion of C-starts and replacing with avoidance behaviour may help compensate for the costs of increasing thermal conditions.

While other studies have shown that elevated temperature affects the locomotor performance in fast-starts (Webb & Zhang, 1994; Beddow, Leeuwen & Johnston, 1995), we found projected increased sea temperatures had no effect on maximum swim speed or escape distances in our species, regardless of exposure duration. Similarly, we found no effect on response latency with temperature and exposure duration. The lack of change in performance at these temperatures indicates once a C-start was initiated, the physical ability to escape is maintained. This suggests future warming may have little effect on mortality rates. Although promising, this may be problematic because metabolic demands for fish increase with rising ambient temperature (Nilsson, Östlund-Nilsson & Munday, 2010; Grenchik, Donelson & Munday, 2013) and C-starts are high energy manoeuvres (Jayne & Lauder, 1993). If an individual’s energy budget is limited, then more resources must be allocated to the rising cost of metabolic demands and potentially leave less available for other activities such as escape behaviour (Careau et al., 2008). Thus, the behavioural modulation of reducing C-start responsiveness after long-term exposure observed, may act as a means of balancing the increasing cost of C-starts at higher temperature.

To conclude, both species showed negative effects of increased temperature after short-term, though the impact was greater for P. amboinensis. After long-term exposure, P. moluccensis adjusted to temperature with beneficial plasticity and energy saving strategies. These results show the C-start escape response possesses thermal plasticity and extended exposure can induce beneficial changes. The difference in response by species matches a previous study showing P. moluccensis possessed a higher capacity for thermal plasticity in aggressive interactions (Warren et al., 2016). Changes in escape behaviour will ultimately affect mortality rates in prey and the species-specific response suggests some will better cope with future temperatures. This study is important as it illustrates the specific contribution that elevated temperature will have on individual performance. However, comprehensive climate change scenarios will also include responses to changes in ocean chemistry (CO2 levels and pH). Furthermore, precise measurement of escape success will require future study with only the prey, only the predator, and both exposed to elevated temperature as predators will also likely have specific responses to thermal conditions. Future studies may also consider a longitudinal approach with several populations from differing sites to consider adaptation to local thermal environments. Studies investigating the factors that influence predator–prey interactions, like this one, are crucial because these interactions determine population sizes and replenishment, leading to changes in community structure.

Supplemental Information

Supplemental Information 1 Dataset for behavioural and kinematic perfromance

Percentages, means, and standard deviations for responsiveness (percent of reactions), directionality (percent of turns), response latency (ms), max speed (body lengths s−1), and response distance (cm).

Click here for additional data file.

We thank Dave Stewart of Affordable Charters, captain of the Kalinda, and the dive volunteers for field assistance. We thank the JCU team at MARFU and Shannon McMahon for their help in the construction and maintenance of the holding facilities. We thank Katie Motson and Shannon McMahon for their aid in collecting burst responses videos.

Additional Information and Declarations

Competing Interests

Author Contributions

Animal Ethics

Field Study Permissions

Data Availability

The authors declare there are no competing interests.

Donald T. Warren conceived and designed the experiments, performed the experiments, analyzed the data, wrote the paper, prepared figures and/or tables, reviewed drafts of the paper.

Jennifer M. Donelson conceived and designed the experiments, performed the experiments, wrote the paper, reviewed drafts of the paper.

Mark I. McCormick conceived and designed the experiments, wrote the paper, reviewed drafts of the paper.

The following information was supplied relating to ethical approvals (i.e., approving body and any reference numbers):

All work reported herein was conducted under permits from James Cook University Animal Ethics Committee (A2079).

The following information was supplied relating to field study approvals (i.e., approving body and any reference numbers):

Collecting permits were approved by the Great Barrier Reef Marine Park Authority (G10/33239.1) and Queensland Fisheries (170251).

The following information was supplied regarding data availability:

The raw data has been supplied as a Supplementary File.

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
