# Peer review of "Extended exposure to elevated temperature affects escape response behaviour in coral reef fishes"

_PeerJ, doi:10.7717/peerj.3652_

## Round 0.1 · original submission · Major Revisions

Please revise your manuscript, taking all of the reviewers' comments into account. Note that your revised paper will be sent out for review again, so it is essential to carefully address each point made by your reviewers.

Reviewer 1 ·

Basic reporting

This manuscript appears to follow the format outlined for your journal. In my opinion, the use of stacked histograms and scales on the graphs are not the best way to communicate their results.

Experimental design

I could not reconcile certain features of the experimental design. The first was the selection of species and the number of different species used. The rationale for these species is provided on lines 92 to 100, but it was not clear why two species were used and its relevance to the goals of the study outlined in lines 79 to 85. This needs to be better explained.

I am also concerned about the scale of the manipulations on the dependent parameter (1 and 2 C). No information was reported on the variation in these parameters. This should be reported so readers have confidence that fish subject to these different treatments were really exposed to different thermal regimes. This becomes particularly important for the longer term results so that readers can have confidence that the result was due to behavioural plasticity and not convergence of the thermal regimes.

Validity of the findings

The manipulations of temperature were put into the climate change scenario outlined in the introduction but the magnitude of these changes gave me several pragmatic concerns. The first was the practicality of such a small experimental manipulation. For there to be a real experimental effect there must be no overlap between the 3 conditions (29, 30 & 31). For this it is important to report the variation in temperature associated with the three conditions and that data was not presented. The second was the rationale to investigate manipulations in climate need to be distinct from seasonal variation, which I assume would lie well within these parameters. How do I know which one I am looking at within this paper, and does it matter? Finally, can you realistically address climate change scenarios from the perspective used in this paper? These experiments address specifics on Mauthner cell function subject to short and long-term exposure to changes in temperature but without a context I once again struggle with understanding what these results mean. Presumably there are other changes that will correlate with changing sea temperature (e.g., pH). These changes in the physical environment will also impact the prey, predators and competitors of the species used in this experiment. While I do appreciate that this becomes an extraordinary complex scenario, my point is that the precise focus used in this paper to investigate one of many antipredator behaviours will limit the insight provided into the impact that climate change will have on predator-prey interactions.

Reviewer 2 ·

Basic reporting

This manuscript is written well and easy to follow. There is sufficient introductory and background material. Figures are clear.

Experimental design

a) The researchers state the length of the fish at the end of the treatment, however, it is necessary to state the average lengths in each group. Did temperature have a difference on growth rate that could account for the minor difference observed? Since the authors presumably have this data it would make an interesting additional figure. Assuming that the weights were not significantly different, the authors could omit a change in eating pattern brought on by elevated temperature explaining the decrease in responsiveness.

b) It should be stated in the methods how often and what the fish were fed during the duration of the experiment.

c) More details about the housing of the fish are necessary. Were they housed in flow-through tanks? Were they housed individually (Damselfish have a tendency to be territorial and aggressive so hopefully this was the case)? How was the temperature maintained and adjusted? How often were temperature measurements made in the holding tanks? What kind of temperature probe was used?

d) Why is temperature measured only in whole numbers and not with three significant digits? The authors should use the same style as Allan et al., 2015 and report to tenths of a degree Celcius.

e) What was the approximate age of the fish? A reader can work backwards and try to figure this out but it would help to simply state this around line 100-101

f)Was the gender of the fish observed? This should be stated even if this was not determined.

g) What was the temperature of the testing arena? Was the water in the arena the same temperature as the experimental group? Was fresh water used for every fish?

Validity of the findings

No comment

Additional comments

This is a very well written and designed study. With a few minor additions to the paper I would strongly recommend this paper for publication.

A few other minor points:

-The authors use the term ‘behavioral decisions’ in the abstract which implies higher order cognitive processing, however this was not tested. Where have the authors tested the decision making ability of the fish? This should be revised to reflect the findings in the manuscript. ‘Behavioral response’ would seem more appropriate.

-Be consistent with ‘behavior’ (ex. in the abstract) and ‘behaviour’ (line 43)

-Line 96 – space needed in the reference

·

Basic reporting

The article is concise and to the point, though a number of important details (though minor changes) regarding experimental design are either missing or should be elaborated on. See general comments.

The data file provided is not the raw data.

Experimental design

The article meets the scope of the journal and the aims are well articulated. The understood absence of a time control is concerning and should be explicitly acknowledged, but not fatal.

Validity of the findings

I make a number of suggestions relating to the analysis, which I believe will better convey the data. The conclusions do appear to be supported by the data.

Additional comments

The paper outlines the effect of elevated temperature on performance of escape responses in two damselfish species. The data appears rigorously collected and the interpretation appropriately supported by the data. The writing on the whole is concise and on point. However, I do think the clarity of the paper could be significantly improved before publication, involving various suggestions relating to each the analysis, and clarifications in the information provided that I believe will make the paper easier to interpret.

A binomial glm(m) would more explicitly describe your data. You can then simultaneously look at the effects of different treatment durations and temperatures, with estimations to describe the importance of each parameter, alike your ANOVA of normally distributed data. This will also yield more informative effect size estimates which could better inform any predictions you wish to make of the effect climate change may have. This is of course not relevant to the 3 level response variable and a multinomial (while possible) is probably more trouble than it is worth. On request I can send sample code for a binomial glm(m) in R.

Please define and describe the “repeated/able drop stimulus”. I am not familiar with this term.

ANOVA output (including effect sizes) should be included so potential meta-analyses can use your work (line 193). If averse to presenting this in the body of the MS, at least throw it into a supplementary table.

The raw data is not provided in the supplementary material, just a summary of averages and proportions. Please add to complement lines 124-5 and show the relative breakdown of sample sizes across treatments.

I would appreciate more information on the ecology of the two species and site (where available).
1) What are the natural temperature ranges of the species, that of the collection site, the range of temperatures from the recruitment sites around your site (ln 99)?
2) Are there large life-history differences between species e.g. longer larval stages which may allow for long migration?
3) How much warming has your site already seen? Ie. Is the baseline ‘control’ historic temperature or just current temperature?
4) How largely distributed are the two species?
5) How separate are the two species taxonomically within the genus?
This could all lead to rather intuitive predictions on their potential to plastically respond to different thermal regimes and the robustness of the metapopulation or species and aid the comparability with the broader ecological literature. Addressing this up front would then allows for a more compelling discussion on perhaps why you see the differences between species (while being careful not to sell ideas as a priori hypotheses).

24: adaptive a better word than beneficial?

51: “changes *to the* phenotype”
Can also lead to chronic stress too. Can setup that different species may react differently.

87: maybe move this to end of experimental methods?
Paragraph starting 92: Personally, I’d like to see this elaborated on and combined with the last paragraph of the introduction.

108: Future studies could improve the power and gain further information by collecting the data longitudinally, with repeated measures through time of individual fish.

115: There may be substantially more micro-environmental variation than AIMS picks up.

112-115: Feeding regime belongs here. Energy budgets could be very important to the likelihood of escape, particularly at elevated temperature of an ectotherm.

124: Do you have Control fish taken at 4-days and 90-days as well? This could be important for controlling for time in captivity effects and should be explicitly acknowledged as a limitation if not.

136: “in *an* attempt”

150-1: On first attempt, or any attempt?

152-3: What constitutes “away”? 180 degrees?

170: Presumably you just mean the random effect for the ANOVA? On what basis was it dropped? AIC or P-value? Seems somewhat superfluous information as you do not present the model.

216-7: Control treatment being natural conditions?

244: I’d suggest the energy demands of more C-starts would be somewhat negligible in comparison to the general increase in maintenance costs of an ectotherm living in higher temperatures. Regardless, the paragraph seems somewhat tangential.

254: I’d suggest breaking this down into a multilevel analyses (inc. data collection) could also be highly beneficial. Some genotypes/individuals will be more responsive (or ‘plastic’) than others, allowing selection to act on plasticity. Plasticity through the larval stage or parental effects could be predictive of reef temperatures and already informing adult phenotypes.

- David Mitchell

---

## Round 0.2 · Minor Revisions

Please take care to address the remaining concerns of the two reviewers.

Reviewer 1 ·

Basic reporting

No comment

Experimental design

No comment

Validity of the findings

See general comments for the author

Additional comments

I have reviewed the revised manuscript and for the sake of brevity am restricting my comments to the revisions addressing my original review.

When I first reviewed this manuscript I did so with too great an expectation regarding the ecological relevance of both the species used and the methods and experimental design employed. The authors have made clear in their response to my comments that the project was much more straightforward and sought to quantify the changes in fast start response associated with thermal change over varying lengths of time.

For that reason I am fine with the revised rationale for the use of species, and clarification on the temperature control afforded by the equipment that they used. However, I am less confident in their interpretation of results and its application to climate change and do believe that the authors have continued to overstate their results. Their disclaimer on their ability to interpret climate change impact is in lines 409 – 412, yet comes at the very end of the Discussion. If it is necessary to reduce the length of the manuscript, it may be worthwhile reducing this section.

·

Basic reporting

The article remains well written. I have nothing substantial to add on the previous review. I have only pedantic points to make before I'd be happy to see the article published.

Experimental design

no comment

Validity of the findings

no comment

Additional comments

23: “affected”

36: may, not will

77: I still don’t think the “drop stimulus” is clear to its meaning. Same in the abstract. It lacks context for the reader to understand. Maybe “we repeatedly produced a fright response/stimulus”, or something to that effect.

88: Move to end of methods

111: Clarify you mean the fish varied in size. Reads slightly awkward and I'm not 100% sure I’m reading it correctly.

204: If you could give a delta AIC, that’d be more proper here. That’s the difference between AIC with and without the term.

Results: The effect sizes are still not displayed, neither in the text nor the supplements. However, the rebuttal letter said they were, so I’m guessing they were forgotten when uploading. They’re useful and necessary to make the paper usable for meta-analytic studies.

207: No significant parameter?

209-210: 84% c-starts, 7% didn’t respond and 7% avoidance behaviour and 2% rounding error, correct? Rephrase slightly as could be read as 7% of the 16% that didn’t produce c-start. Same with

227. Also, stay consistent with number of significant figures.

245-250, 294-295: The real advantage of suggesting the binomial model was that you could explicitly test whether the sensitivity of the two species to temperature. I.e. if temperature and species interacted. Would also allow you to control for size or any other factor, where the contingency analyses are limited. Familiarising yourself with such methods will be greatly beneficial if continuing with "repeated measures" work.

If the authors are persisting with this sort of work, I’d suggest they give some thought to the ideas of Bayesian updating in plasticity, see Stamps & Franenhuis (2016 TREE, and citations within) on incorporating information sequentially in the plastic response. And empirical work by Stephan Munch and Anurag Agrawal looking at how different type of plasticity may interact (e.g. transgenerational plasticity, developmental, etc.). This could lead to some interesting predictions on how your populations might plastically respond to a changing climate.

- David Mitchell

---

## Round 0.3 · accepted · Accept

Thank you for making the additional revisions.